# High Tumor Burden Predicts Poor Response to Enzalutamide in Metastatic Castration-Resistant Prostate Cancer Patients

**DOI:** 10.3390/cancers13163966

**Published:** 2021-08-05

**Authors:** Yu-Ting Hsieh, Bing-Juin Chiang, Chia-Chang Wu, Chun-Hou Liao, Chia-Da Lin, Chung-Hsin Chen

**Affiliations:** 1Department of Urology, National Taiwan University Hospital, Taipei 10002, Taiwan; ag1233044@gmail.com; 2Division of Urology, Department of Surgery, Cardinal Tien Hospital and School of Medicine, Fu-Jen Catholic University, New Taipei City 23148, Taiwan; bingjuinchiang@gmail.com (B.-J.C.); charleswjj@yahoo.com.tw (C.-H.L.); 3Department of Urology, Shuang Ho Hospital, Taipei Medical University, New Taipei City 23561, Taiwan; liaoch22@gmail.com (C.-C.W.); 20500@s.tmu.edu.tw (C.-D.L.); 4Department of Urology, School of Medicine, College of Medicine, Taipei Medical University, Taipei 11031, Taiwan; 5TMU Research Center of Urology and Kidney (TMU-RCUK), Taipei Medical University, Taipei 11031, Taiwan

**Keywords:** prognosis, prostatic neoplasms, enzalutamide, prostate-specific antigen, tumor burden

## Abstract

**Simple Summary:**

Several available medications are involved in the treatment of metastatic castration-resistant prostate cancer (mCRPC). In this Taiwanese mCRPC cohort receiving enzalutamide patients with high tumor burden (HTB) were defined as those with either appendicular bony metastasis or visceral metastasis. A high tumor burden reduced the PSA response rate, radiological response rate, and progression-free survival duration. In addition, patients’ comorbidities and laboratory data—such as ALP, LDH, ALT, and hemoglobin—were correlated with the treatment efficacy of enzalutamide. Our study revealed a tumor burden before the use of enzalutamide was associated with treatment outcomes. The physician can use this information to estimate the response rate of enzalutamide and help formulate a personalized treatment plan for mCRPC patients.

**Abstract:**

To assess the predictive value of tumor burden on the biochemical response, and radiological response in Taiwanese metastatic castration-resistant prostate cancer (mCRPC) patients receiving enzalutamide. The mCRPC patients treated with enzalutamide were recruited from three hospitals. High tumor burden (HTB) was classified as metastases at either appendicular bone or visceral organ. Good prostate-specific antigen (PSA) response was defined as PSA reduction of 80%. In this cohort, there were 104 (54.2%) HTB patients and 88 (45.8%) with low tumor burden (LTB). Compared to LTB patients, fewer HTB patients had good PSA response (odds ratio: 0.43, range: 0.22–0.87, *p* = 0.019) and fewer radiological response (complete and partial remission) (odds ratio: 0.78, range: 0.36–1.68, *p* = 0.52) to enzalutamide. The disease control rate which also contained stable disease, was still lower in HTB (76.0%) than LTB group (92.9%, OR: 0.24, range: 0.07–0.77, *p* = 0.016) in the multivariable model. In addition, HTB patients had significantly shorter progression–free survival duration than did LTB patients (median: 8.3 vs. 21.6 months, log-rank test *p* = 0.003) in the univariable analysis. The tumor burden before the use of enzalutamide was associated with treatment outcomes. HTB reduced PSA response rate, radiological response rate and progression-free survival duration.

## 1. Introduction

The incidence and mortality of prostate cancer are rising in several Asian countries [1]. In Taiwan, the incidence of prostate cancer is also increasing and is now more than 30 cases per 100,000 person-years [2]. In Taiwan among newly-diagnosed prostate cancer cases, over 25% have metastatic disease [2]. Although most prostate cancer patients do not die from their prostate cancer, in the case of those with metastatic prostate cancer (mPC), death is cancer-related and their median overall survival is between 33 months and 45 months [3,4]. In addition, the sites of prostate cancer metastases have a significant impact on mPC patients’ outcomes. An Asian study series also reported that cases of mPC with high volume disease had poorer overall and cancer-specific survival [3].

The chemohormonal androgen ablation randomized (CHAARTED) trial effectively highlighted the impact of the mPC disease burden and demonstrated that the upfront use of docetaxel only benefited the metastatic castration-naive prostate cancer (mCNPC) patients with high volume disease (HVD), but not those with low volume disease (LVD) [5,6]. Furthermore, the LATITUDE and STAMPEDE trials also proved the upfront use of abiraterone helps improve progression-free and overall survival in mCNPC patients with a high tumor burden [7,8]. Therefore, in patients with mCNPC, the tumor burden is an important clinical factor in deciding on an appropriate treatment strategy. 

Nevertheless, information is lacking regarding the impact of the tumor burden on the treatment response in those mPC patients who are already refractory to androgen deprivation therapy (ADT). Based on the current treatment guidelines, enzalutamide is one of the preferred first line medications for metastatic castration refractory prostate cancer (mCRPC) [9]. Enzalutamide was shown to increase mCRPC patients’ median overall survival duration to 2.2 months in the pre-docetaxel (PREVAIL) trial and to 4.8 months in the post-docetaxel (AFFIRM) trial [10,11]. However, these two pivotal studies did not analyze the influence of tumor volume on the response and survival outcomes.

The goal of the present study was to assess the predictive and prognostic role of the tumor burden in mCRPC patients receiving enzalutamide. To do so, we used a collaborated cohort study from three hospitals to investigate the impact of tumor burden on the PSA response, the radiological response rate, and progression-free survival duration in mCRPC patients.

## 2. Materials and Methods

### 2.1. Patient Population and Selection Criteria

Between February 2020 and October 2020, consecutive patients with mCRPC who received new hormonal agents, such as enzalutamide and abiraterone, at the National Taiwan University Hospital, the Taipei Medical University-Shuang Ho Hospital, and the Cardinal Tien Hospital, were enrolled in this study. The patients who ever used abiraterone prior to enzalutamide was excluded. The subjects without complete information for the outcomes and were excluded as well. (Figure 1) The study was reviewed and approved by the local institutional review board (201305059RINC, NTUH; N20201013 TMU; CTH-109-3-5-051).

### 2.2. Definition of Tumor Burden

Tumor burden was defined by the existence of appendicular bony metastasis and visceral metastasis prior to the administration of enzalutamide. Patients with a high tumor burden (HTB) were defined as those with either appendicular bony metastasis or visceral metastasis [12]. The poor outcome of the patients with appendicular bony metastasis has been validated [13]. The remaining patients were considered to be in the low tumor burden (LTB) group.

### 2.3. Clinical Information Collection and Outcome Measurement

Clinical information was obtained from medical records using a standardized questionnaire (by YTH) and critically reviewed (by CHC). Data included: date of diagnosis, diagnosis age, initial symptoms, Gleason score, clinical stages, initial and all sequential treatments, date of castration, mPC and mCRPC treatment dates, and date of (and reasons for) death. In addition, Eastern Cooperative Oncology Group performance status, laboratory data, skeletal-related events, and tumor burden before the use of enzalutamide were obtained. We investigated three different important outcomes to evaluate the impact of enzalutamide on tumor burden. They were as follows: (1) The PSA response was defined as the maximal fall in PSA percentage after the use of enzalutamide. A PSA drop greater than 80% was designated as a good PSA response. (2) The radiological response was determined using the response evaluation criteria in solid tumors (RECIST) to assess changes in tumor burden. (3) Progression-free survival duration was defined as the duration between the start date of enzalutamide treatment and the date of PSA or radiological progression. PSA progression was determined based on the following criteria: (1) the PSA increased by ≥25% and ≥5 ng/mL in cases where the PSA did not decrease after the use of enzalutamide; (2) PSA increased ≥25% and ≥5 ng/mL in cases where the PSA decreased ≤50% after the use of enzalutamide; (3) PSA increased ≥50% and ≥5 ng/mL in cases where the PSA decreased ≥50% after the use of enzalutamide [14].

### 2.4. Statistical Analysis

All statistical analyses were tabulated and analyzed by Stata/MP version 16. The Chi-square test was used to evaluate the association between categorical variables and the Mann–Whitney U test was applied to compare the continuous variables. The Kaplan–Meier method with log-rank test and Cox proportional hazard models were utilized to compare progression–free survival duration among the patient groups. The logistic regression models were used to compare PSA and radiological response among the patient groups. All the above models were based on the patients with available information on outcomes, such as progression-free survival duration (N = 192), PSA response (N = 146), and radiological response (N = 131). The factors with statistical significance (*p* < 0.1 in univariable analysis) were selected for the multivariable model. All tests were two-sided with *p* < 0.05 considered statistically significant. The missing value would not be included in the Cox proportional analysis.

## 3. Results

### 3.1. Patient Demographics

A total of 192 mCRPC patients were enrolled in this study (Table 1). The median age at commencement of enzalutamide was 72 years (range, 46–94 years). HTB diseases were identified in 104 (54.2%) patients, and the remaining 88 (45.8%) patients were classified as having LTB disease. The patients with HTB were younger (median: 71 (range, 46–90) vs. 74 (range, 51–94) years, *p* = 0.065) and had higher PSA values at diagnosis (median: 256.6 (range, 3.2–16,410) vs. 63.4 (range, 3.3–9424.7) ng/mL, *p* < 0.001) than did those with LTB. The HTB patients had a significantly higher proportion of bone pain (21.2% vs. 6.8%), skeletal-related events (18.3% vs. 6.8%), and prior chemotherapy history (41.4% vs. 21.6%) compared to the LTB patients. The PSA value of more than 50 ng/mL before enzalutamide usage was significantly more frequently noted in HTB than in LTB patients (33.7% vs. 15.9%, *p* = 0.004). The chemotherapeutic agent used prior to enzalutamide was docetaxel in Taiwan. None of our patients received cabazitaxel before the administration of enzalutamide. Other demographic factors—including Gleason sum, ECOG performance status, and so on—were similar between the groups.

### 3.2. PSA Response

We considered a good PSA response to be the reduction in PSA value of greater than 80% after the use of enzalutamide. The HTB patients had fewer good PSA response rates than did the LTB patients (odds ratio (OR) for HTB: 0.43; range, 0.22–0.87, *p* = 0.019). Among the HTB patients, 53%, 16%, and 17% had a PSA reduction of ≥80%, ≥50%, and <50%, respectively. By contrast, 72%, 15%, and 11% of LTB patients experienced PSA reductions of ≥80%, ≥50%, and <50%, respectively (Figure 2). In the univariable analysis for good PSA response, initial presentation with both abnormal PSA and DRE (OR: 2.48, range, 1.11–5.58, *p* = 0.028), hypertension (OR: 2.1, range: 1.06–4.18, *p* = 0.033), and LDH value (OR: 0.2, range, 0.04–0.94, *p* = 0.041) were statistically significant factors. The multivariable analysis consisted of significant factors in the univariable analysis (*p* < 0.1), and revealed that hypertension (OR: 3.0, range, 0.31–6.84, *p* = 0.05) and a hemoglobin value of more than 12 mg/dL (OR: 2.7, range, 1.07–6.8, *p* = 0.035) significantly increased a good PSA response rate. In addition, tumor burden was also a statistically significant factor for predicting PSA responses. The LTB patients had more good PSA response rates than the patients with HTB (OR: 0.44, range, 0.19–1.0, *p* = 0.05) (Table 2).

### 3.3. Radiological Response

The HTB patients had significantly fewer rates of partial radiological response and stable disease than did the LTB patients (Figure 3). The disease control rate, including the combination of complete response, partial response, and stable disease, was significantly lower in the HTB (76.0%) than LTB group (92.9%, OR: 0.24, range, 0.07–0.77, *p* = 0.016). The multivariable analysis for the disease control showed that HTB (OR: 0.21, range, 0.06–0.77, *p* = 0.018) predicted a poor response, whereas a higher hemoglobin (OR: 4.14, range, 1.0–1.0, *p* = 0.043) reflected good disease control using enzalutamide (Table 3). 

### 3.4. Progression-Free Survival Duration

The HTB patients had a significantly shorter progression–free survival duration than did the LTB patients (median: 8.3 months, 95% confidence interval (CI): 5.8–10.8 vs. 21.6 months, 95% CI: 12.1–31.2) (log-rank test, *p* = 0.003) (Figure 4). In addition to tumor burden, several characteristics, including the duration from mPC to mCRPC, PSA value before using enzalutamide, ECOG performance status, prior chemotherapy history, skeletal-related events, and some laboratory data (hemoglobin, percent of segment neutrophil, platelets count, alanine aminotransferase (ALT), alkaline phosphatase (ALP), lactic dehydrogenase (LDH), and calcium) were associated with progression-free survival in the univariable analysis. The multivariable analysis revealed higher PSA value before using enzalutamide (hazard ratio (HR): 2.43 for PSA ≥50 ng/mL vs. PSA < 10 ng/mL; *p* = 0.008), prior chemotherapy history (HR: 2.23, *p* = 0.021), higher percent of segmental neutrophil (HR: 2.18, *p* = 0.033), higher ALT (HR: 0.53, *p* = 0.023), higher ALP (HR: 2.18, *p* = 0.051), and higher LDH (HR: 7.46, *p* < 0.001) significantly reduced progression-free survival duration. However, tumor burden lost its significance in predicting progression-free survival duration in the multivariable analysis (HR: 1.40, *p* = 0.221) (Table 4).

## 4. Discussion

In this mCRPC cohort receiving enzalutamide, we identified tumor burden as a significant factor in predicting the effect of treatment, including the PSA response and the radiological response. The physician can use this finding to estimate the response rate of enzalutamide and help formulate a personalized treatment plan for mCRPC patients.

In several clinical trials, the extensiveness of tumors has been typically (and logically) regarded as a prognostic factor and treated as a stratified criterion. For example, the CHAARTED trial defined high volume disease as the presence of visceral metastases and/or ≥4 bone lesions with ≥1 lesion beyond the vertebral bodies and pelvis. This study revealed the survival benefit of the upfront use of docetaxel only existed in mPC patients with high volume disease but not in those with low volume disease [5]. In an Asian series of metastatic castration-sensitive prostate cancer patients, the tumor volume was significantly associated with overall survival (OS) and cancer-specific survival (CSS) [3]. Although the LATTITUDE study used a somewhat different definition of tumor burden, it still showed the value of tumor burden in selecting those cases who would receive a survival benefit with the early use of abiraterone [8,15]. In our study, in addition to being a stratified factor in clinical trials, tumor burden was noted to be a significant factor in predicting treatment response.

The PSA value is a direct indicator of treatment response and is correlated with OS and progression-free survival [16]. The CHAARTED trial revealed that patients whose PSA nadir was less than 0.2 ng/mL had a significantly longer OS duration than did those with a PSA nadir >4 ng/mL. In addition, more patients who were treated with ADT plus docetaxel had PSA nadir <0.2 ng/mL compared to those managed with ADT alone [5]. The analyses from the COU-AA-301 and COU-AA-302 trials also showed that PSA kinetics, including PSA nadir, PSA response, PSA doubling time, and time-to-PSA progression, were highly correlated with OS [17,18]. Moreover, early PSA response was noticed to be significantly associated with good survival outcomes in mCRPC patients treated with either abiraterone, enzalutamide, or ketoconazole [19]. All pivotal clinical trials, such as the ARCHES, ENZAMET, and LATTITUDE trials, lacked a detailed description of PSA response between HVD and LVD [8,20,21]. Our study provided this missing element by showing that a higher PSA response rate was noted in those mCRPC patients with LTB compared to those with HTB.

A post-hoc analysis of the Asian PREVAIL trial indicated that, in Japanese, Korean, and East Asian mCRPC patients treated with enzalutamide, the percentage of patients who achieved a PSA reduction of ≥50% from baseline was 60.7%, 70.0%, and 68.5%, respectively [22,23,24]. The treatment effects of enzalutamide in terms of radiographic progression-free survival, OS, and time-to-PSA progression were consistent with those observed in the overall study population of the PREVAIL trial [25]. Our study of Taiwanese mCRPC patients revealed the rate of PSA reduction ≥50% from baseline was slightly higher (77.4%) than that in PREVAIL and Asian PREVAIL trials. A possible reason was that our cohort had mildly lower Gleason scores and pretreatment PSA values than those patients in the PREVAIL and Asian PREVAIL trials.

Several laboratory biomarkers have been shown to be associated with the outcomes of mCRPC patients treated with either enzalutamide or abiraterone [26]. In the AFFIRM trial, pretreatment ALP and LDH levels were correlated with PSA value [27]. Moreover, ALP level prior to enzalutamide treatment has been proven to be a strong predictor of progression-free survival in a real-world cohort [28]. Furthermore, it has been observed in several mCRPC series that the higher the LDH value before treatment with enzalutamide or abiraterone, the poorer the progression-free survival [29,30,31]. Our findings regarding ALP and LDH in our Taiwanese cohort were consistent with those of other internationally previously published series.

Enzalutamide has been approved as a standard first-line therapeutic agent for mCRPC by the FDA, but some patients have not benefited from it due to tumor heterogeneity, genetic background, ethnicity, and/or other reasons [32]. In addition, most studies have compared outcomes between therapeutic agents, and few have focused on the comparison of a single drug in different types of patients. Our study identified a relationship between tumor burden and prostate cancer patients treated with enzalutamide in a Taiwanese cohort. Additionally, we derived some useful findings, such as that the duration of time from mPC to mCRPC and prior chemotherapy could predict progression-free survival duration. Further research may establish a useful model to develop optimal therapeutic decisions.

Our study has several limitations. Firstly, the case number was not enough to assess all clinical parameters in the multivariable analysis. Nevertheless, ours was the largest comparable cohort in Taiwan studied thus far, and it served to prove our hypothesis. Secondly, this was a retrospective analysis; however, selection bias may have been partially mitigated by recruiting consecutive mPC patients. Thirdly, the variety of treatments used in different institutions may interfere with study analysis; however, most of the patients in our cohort received enzalutamide reimbursed by the national health system, which would review the related medical records in advance. Therefore, there would be minimal variation of treatment plan. Finally, the administration of enzalutamide was reimbursed and should follow the regulation of the national health system in Taiwan. The patients who had duration of less than 12 months from mPC to mCRPC were suggested to have chemotherapy instead of enzalutamide. Therefore, the regulation limited us to enroll patients and analyze the effect of tumor burden for the patients with poor response to initial hormonal therapy.

## 5. Conclusions

In conclusion, our study revealed a tumor burden before the use of enzalutamide was associated with treatment outcomes. A high tumor burden reduced the PSA response rate, radiological response rate, and progression-free survival duration. In addition, patients’ comorbidities and laboratory data—such as ALP, LDH, ALT, and hemoglobin—were correlated with the treatment efficacy of enzalutamide.

## Figures and Tables

**Figure 1 cancers-13-03966-f001:**
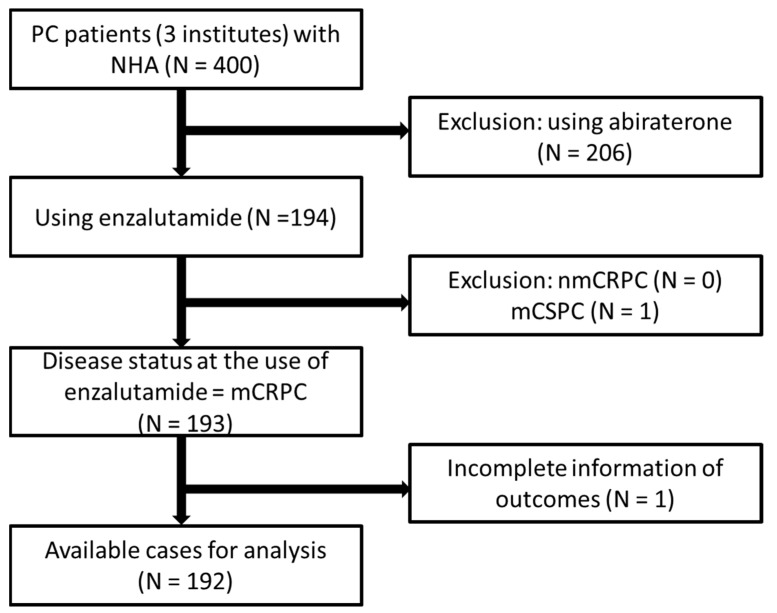
Patient selection flow diagram PC = prostate cancer; NHA = new hormonal agents including abiraterone acetate and enzalutamide; mCRPC = metastatic castration resistant prostate cancer; nmCRPC = non-mCRPC; mCSPC = metastatic castration sensitive prostate cancer.

**Figure 2 cancers-13-03966-f002:**
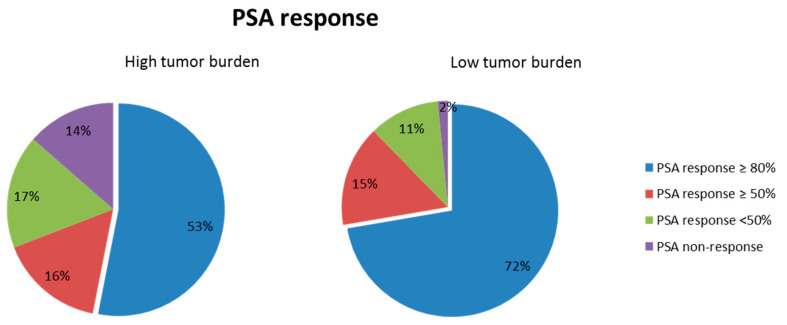
PSA response in the patients stratified tumor burden. PSA response defined the maximal PSA drop percentage after the use of enzalutamide. PSA drop more than 80% was referred as the good PSA response.

**Figure 3 cancers-13-03966-f003:**
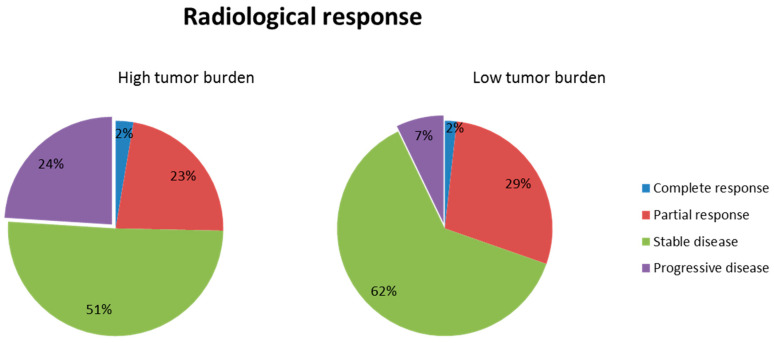
Radiological response in the patients stratified tumor burden.

**Figure 4 cancers-13-03966-f004:**
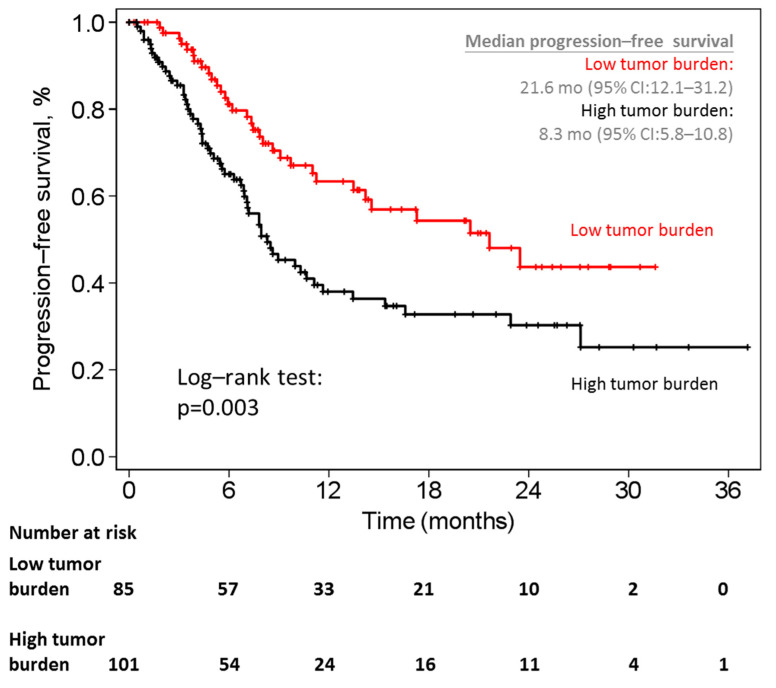
Kaplan–Meier curves of progression-free survival between patients with high tumor burden and low tumor burden. mo = months; CI = confidence interval.

**Table 1 cancers-13-03966-t001:** Demographics of prostate cancer patients stratified by tumor burden.

Groups *	All Patients	Low Tumor Burden		High Tumor Burden		*p*-Value
Patient number (n)	192	88	45.8%	104	54.2%	
Median age (range, years)	72 (46–94)	74 (51–94)		71 (46–90)		0.065
Median mPC to mCRPC (range, months)	14 (0–148.9)	13.3 (0–148.9)		14.3 (0–125.9)		0.955
Median mCRPC to enzalutamide (range, months)	4 (0–106.2)	4 (0–72.1)		3.9 (0–106.2)		0.848
Median PSA at diagnosis	117.3 (3.2–16,410.5)	63.4 (3.3–9424.7)		256.6 (3.2–16,410)		<0.001
ECOG						0.278
0	105	53	60.2%	52	50.0%	
1	67	29	33.0%	38	36.5%	
2 and 3	20	6	6.8%	14	13.5%	
Gleason sum						0.724
≤6	10	5	5.7%	5	4.8%	
3 + 4	12	4	4.5%	8	7.7%	
4 + 3	29	15	17.0%	14	13.5%	
8~10	128	57	64.8%	71	68.3%	
Missing	13	7	8.0%	6	5.8%	
Abnormal PSA only						0.368
Yes	71	36	40.9%	35	33.7%	
No	121	52	59.1%	69	66.3%	
Abnormal PSA and DRE						0.171
Yes	54	29	33.0%	25	24.0%	
No	138	59	67.0%	79	76.0%	
LUTS						0.664
Yes	90	43	48.9%	47	45.2%	
No	102	45	51.1%	57	54.8%	
Hematuria						0.473
Yes	8	5	5.7%	3	2.9%	
No	184	83	94.3%	101	54.9%	
Bone pain						0.007
Yes	28	6	6.8%	22	21.2%	
No	164	82	93.2%	82	78.9%	
Other metastasis-related symptoms						0.064
Yes	5	0	0.0%	5	4.8%	
No	187	88	100%	99	95.2%	
Skeletal-related events						0.03
Yes	25	6	6.8%	19	18.3%	
No	167	82	93.2%	85	81.7%	
Prior chemotherapy						0.005
Yes	62	19	21.6%	43	41.4%	
No	130	69	78.4%	61	58.7%	
PSA response						0.026
>80%	90	47	53.4%	43	41.3%	
<80%	56	18	20.5%	38	36.5%	
NA or missing	46	23	26.1%	23	22.1%	
Radiological response						0.017
CR, PR and SD	109	52	59.1%	57	54.8%	
PD	22	4	4.5%	18	17.3%	
NA or missing	61	32	36.4%	29	27.9%	

* High tumor burden was defined as those with either appendicular bony metastasis or visceral metastasis. mPC = metastatic prostate cancer; mCRPC = metastatic castration-resistant prostate cancer; LUTS = low urinary tract symptoms; NA = not available; CR = complete remission; PR = partial remission; SD = stable disease; PD = progressive disease.

**Table 2 cancers-13-03966-t002:** Univariable and multivariable analyses of PSA response to enzalutamide in metastatic castration-resistant prostate cancer patients.

Variables	Case Number	Events	Univariable			Multivariable *		
			OR	Range	*p* Value	OR	Range	*p* Value
Age	146	90	1.02	0.99–1.06	0.232	—	—	—
Time from mPC to mCRPC (months)	141 (5 missing)	87	1.01	0.99–1.03	0.277	—	—	—
Time from mCRPC to enzalutamide (range, months)	141 (5 missing)	87	1.01	0.98–1.03	0.648	—	—	—
PSA before enzalutamide (ng/mL)								
<10	54	35	1	—	—	—	—	—
10~<50	54	31	0.73	0.34–1.59	0.43	—	—	—
≥50	38	24	0.93	0.39–2.21	0.87	—	—	—
Gleason sum								
≤7	35	24	1	—	—	—	—	—
8~10	100	60	0.69	0.30–1.56	0.369	—	—	—
Missing	11	6	0.55	0.14–2.2	0.397	—	—	—
ECOG								
0	75	46	1	—	—	—	—	—
1	54	32	0.92	0.45–1.87	0.812	—	—	—
2 and 3	17	12	1.51	0.48–4.74	0.477	—	—	—
**Initial presentation**								
Abnormal PSA only								
No	97	59	1	—	—	—	—	—
Yes	49	31	1.11	0.55–2.26	0.775	—	—	—
Abnormal PSA and DRE								
No	104	58	1	—	—	—	—	—
Yes	42	32	2.48	1.11–5.58	0.028	—	—	—
LUTS								
No	78	46	1	—	—	—	—	—
Yes	68	44	1.28	0.65–2.5	0.478	—	—	—
Hematuria								
No	143	89	1	—	—	—	—	—
Yes	3	1	0.30	0.03–3.43	0.335	—	—	—
Bone pain								
No	125	77	1	—	—	—	—	—
Yes	21	13	1.01	0.39–2.62	0.979	—	—	—
Other metastasis-related symptoms								
No	142	87	1	—	—	—	—	—
Yes	4	3	1.90	0.19–18.7	0.584	—	—	—
**Comorbidities**								
Hypertension								
No	57	29	1	—	—	1	—	—
Yes	89	61	2.10	1.06–4.18	0.033	3.00	1.31–6.84	0.009
Diabetes mellitus								
No	111	66	1	—	—	—	—	—
Yes	35	24	1.49	0.66–3.34	0.335	—	—	—
Heart disease								
No	112	68	1	—	—	—	—	—
Yes	34	22	1.19	0.53–2.64	0.675	—	—	—
Liver disease								
No	131	84	1	—	—	1	—	—
Yes	15	6	0.37	0.13–1.11	0.077	0.44	0.12–1.63	0.217
Renal disease								
No	128	78	1	—	—	—	—	—
Yes	18	12	1.28	0.45–3.64	0.64	—	—	—
Symptomatic BPE								
No	92	52	1	—	—	1	—	—
Yes	54	38	1.83	0.89–3.73	0.099	1.52	0.65–3.54	0.332
**Tumor-associated status before enzalutamide**								
Tumor burden								
Low	65	47	1	—	—	1	—	—
High	81	43	0.43	0.22–0.87	0.019	0.44	0.19–1.0	0.05
Prior chemotherapy history								
No	101	67	1	—	—	1	—	—
Yes	45	23	0.53	0.26–1.09	0.082	0.51	0.19–1.35	0.174
Bone pain								
No	120	72	1	—	—	—	—	—
Yes	26	18	1.50	0.60–3.72	0.382	—	—	—
Skeletal-related events								
No	130	80	1	—	—	—	—	—
Yes	16	10	1.04	0.36–3.04	0.941	—	—	—
**Laboratory data before enzalutamide**								
Hemoglobulin (g/dL)								
<12	52	27	1	—	—	—	—	—
≥12	50	35	2.16	0.96–4.87	0.063	2.70	1.07–6.8	0.035
Missing	44	28	1.62	0.71–3.68	0.249	1.89	0.59–6.03	0.283
Platelet count (k/μL)								
<200	55	39	1	—	—	1	—	—
≥200	73	41	0.53	0.25–1.11	0.09	0.48	0.2–1.12	0.088
Missing	18	10	0.51	0.17–1.54	0.233	0.59	0.14–2.56	0.482
LDH (IU/L)								
<200	18	15	1	—	—	1	—	—
≥200	18	9	0.20	0.04–0.94	0.041	0.19	0.34–1.04	0.055
Missing	110	66	0.30	0.08–1.1	0.069	0.16	0.04–0.69	0.014

* Multivariable analysis based on univariable *p* value < 0.1. mCRPC = metastatic castration refractory prostate cancer; mPC = metastatic prostate cancer; OR = odds ratio; DRE = digital rectal examination; LUTS = low urinary tract symptoms; BPE = benign prostate enlargement; LDH = lactate dehydrogenase. Several insignificant laboratory data, such as white blood cells, renal function, liver function, alkaline phosphatase, bilirubin, potassium and calcium, were not shown above.

**Table 3 cancers-13-03966-t003:** Univariable and multivariable analyses of radiological response to enzalutamide in metastatic castration-resistant prostate cancer patient.

Variables	Case Number	Events	Univariable			Multivariable *		
			OR	Range	*p*-Value	OR	Range	*p*-Value
Age	131	109	1.01	0.96–1.07	0.653	—	—	—
time from mPC to mCRPC (months)	128 (3 missing)	108	1.00	0.98–1.03	0.912	—	—	—
Time from mCRPC to enzalutamide (range, months)	127 (4 missing)	107	0.98	0.95–1.01	0.206	—	—	—
PSA before enzalutamide (ng/mL)								
<10	48	39	1	—	—	—	—	—
10~<50	51	44	1.45	0.49–4.26	0.499	—	—	—
≥50	32	26	1.00	0.32–3.15	1	—	—	—
Gleason sum								
≤7	27	21	1	—	—	—	—	—
8~10	99	83	1.48	0.52–4.25	0.464	—	—	—
Missing	5	5	Drop due to perfect prediction			—	—	—
ECOG								
0	72	61	1	—	—	—	—	—
1	46	38	0.86	0.32–2.32	0.761	—	—	—
2 and 3	13	10	0.60	0.14–2.54	0.489	—	—	—
**Initial presentation**								
Abnormal PSA only								
No	87	74	1	—	—	—	—	—
Yes	44	35	0.68	0.27–1.75	0.427	—	—	—
Abnormal PSA and DRE								
No	91	74	1	—	—	—	—	—
Yes	40	35	1.51	0.51–4.46	0.453	—	—	—
LUTS								
No	77	63	1	—	—	—	—	—
Yes	54	46	1.28	0.5–3.3	0.612	—	—	—
Hematuria								
No	128	108	1	—	—	1	—	—
Yes	3	1	0.09	0.01–1.07	0.057	0.21	0.01–4.43	0.319
Bone pain								
No	113	94	1	—	—	—	—	—
Yes	18	15	1.01	0.26–3.84	0.988	—	—	—
Other metastasis-related symptoms								
No	128	107	1	—	—	—	—	—
Yes	3	2	0.39	0.34–4.53	0.454	—	—	—
**Comorbidities**								
Hypertension								
No	50	39	1	—	—	—	—	—
Yes	81	70	1.79	0.71–4.52	0.214	—	—	—
Diabetes mellitus								
No	99	84	1	—	—	—	—	—
Yes	32	25	0.64	0.23–1.74	0.379	—	—	—
Heart disease								
No	102	85	1	—	—	—	—	—
Yes	29	24	0.96	0.32–2.87	0.942	—	—	—
Liver disease								
No	116	96	1	—	—	—	—	—
Yes	15	13	1.35	0.28–6.48	0.704	—	—	—
Renal disease								
No	115	95	1	—	—	—	—	—
Yes	16	14	1.47	0.31–7.0	0.626	—	—	—
Symptomatic BPE								
No	79	67	1	—	—	—	—	—
Yes	52	42	0.75	0.3–1.89	0.546	—	—	—
**Tumor-associated status before enzalutamide**								
Tumor burden								
Low	50	66	1	—	—	1	—	—
High	81	43	0.24	0.07–0.77	0.016	0.21	0.06–0.77	0.018
Prior chemotherapy history								
No	86	75	1	—	—	1	—	—
Yes	45	34	0.45	0.18–1.15	0.095	0.60	0.18–2.00	0.407
Bone pain								
No	109	90	1	—	—	—	—	—
Yes	22	19	1.34	0.36–4.98	0.665	—	—	—
Skeletal-related events								
No	113	95	1	—	—	—	—	—
Yes	18	14	0.66	0.2–2.25	0.509	—	—	—
**Laboratory data before enzalutamide**								
Hemoglobulin (g/dL)								
<12	44	34	1	—	—	1	—	—
≥12	47	43	3.16	0.91–10.7	0.07	4.14	1.04–16.4	0.043
Missing	40	32	1.18	0.41–33.5	0.761	0.56	0.13–2.52	0.454
Segment WBC (%)								
<60	32	26	1	—	—	1	—	—
≥60	41	29	0.56	0.18–1.7	0.304	0.80	0.10–6.24	0.828
Missing	58	54	3.12	0.81–12.0	0.099	4.24	0.33–54.2	0.266
Lymphocyte (%)								
<30	51	39	1	—	—	1	—	—
≥30	43	35	1.35	0.49–3.68	0.562	2.07	0.27–15.7	0.480
Missing	37	35	5.38	1.13–25.8	0.035	2.34	0.24–23.4	0.469

* Multivariable analysis based on univariable *p* value < 0.1. mCRPC = metastatic castration refractory prostate cancer; mPC = metastatic prostate cancer; OR = odds ratio; DRE = digital rectal examination; LUTS = low urinary tract symptoms; BPE = benign prostate enlargement. Several insignificant laboratory data, such as white blood cells, platelet count, renal function, liver function, alkaline phosphatase, bilirubin, potassium and calcium, were not shown above.

**Table 4 cancers-13-03966-t004:** Univariable and multivariable analyses of progression-free survival duration to enzalutamide in metastatic castration-resistant prostate cancer patients.

Variables	Case Number	Failure Events	Univariable			Multivariable		
			HR	Range	*p*-Value	HR	Range	*p*-Value
Age	192	89	1.00	0.98–1.02	0.926	—	—	—
time from mPC to mCRPC (months)	181 (11 missing)	87	0.98	0.97–1.00	0.010	0.99	0.97–1.00	0.058
Time from mCRPC to enzalutamide (range, months)	179 (13 missing)	86	1.00	0.98–1.01	0.676	—	—	—
PSA before enzalutamide (ng/mL)								
<10	79	25	1	—	—	1	—	—
10~<50	64	35	1.99	1.19–3.32	0.009	1.84	0.99–3.40	0.052
≥50	49	29	3.22	1.87–5.55	<0.001	2.43	1.26–4.67	0.008
Gleason sum								
≤7	51	18	1	—	—	1	—	—
8~10	128	66	1.60	0.95–2.70	0.076	1.06	0.59–1.89	0.857
Missing	13	5	12.6	0.47–3.40	0.645	0.93	0.31–2.77	0.898
ECOG								
0	105	41	1	—	—	1	—	—
1	67	38	2.05	1.32–3.20	0.001	1.05	0.59–1.89	0.858
2, 3	20	10	2.35	1.17–4.73	0.017	1.59	0.68–3.77	0.287
**Initial presentation**								
Abnormal PSA only								
No	121	63	1	—	—	—	—	—
Yes	71	26	0.80	0.51–1.27	0.351	—	—	—
Abnormal PSA and DRE								
No	138	63	1	—	—	—	—	—
Yes	54	26	0.84	0.53–1.34	0.466	—	—	—
LUTS								
No	102	47	1	—	—	—	—	—
Yes	90	42	1.03	0.68–1.57	0.872	—	—	—
Hematuria								
No	184	86	1	—	—	—	—	—
Yes	8	3	1.88	0.59–6.01	0.288	—	—	—
Bone pain								
No	164	79	1	—	—	—	—	—
Yes	28	10	0.79	0.41–1.52	0.474	—	—	—
Other metastasis-related symptoms								
No	187	86	1	—	—	—	—	—
Yes	5	3	1.33	0.42–4.22	0.624	—	—	—
**Comorbidities**								
Hypertension								
No	72	38	1	—	—	—	—	—
Yes	120	51	0.72	0.47–1.1	0.129	—	—	—
Diabetes mellitus								
No	146	67	1	—	—	—	—	—
Yes	46	22	1.13	0.7–1.83	0.614	—	—	—
Heart disease								
No	143	67	1	—	—	—	—	—
Yes	49	22	1.06	0.65–1.72	0.819	—	—	—
Liver disease								
No	170	79	1	—	—	—	—	—
Yes	22	10	1.16	00.6–2.24	0.657	—	—	—
Renal disease								
No	171	77	1	—	—	—	—	—
Yes	21	12	1.17	0.63–2.14	0.622	—	—	—
Symptomatic BPE								
No	116	57	1	—	—	—	—	—
Yes	76	32	0.81	0.53–1.25	0.339	—	—	—
**Tumor-associated status before enzalutatmide**								
Tumor burden								
Low	88	32	1	—	—	1	—	—
High	104	57	1.91	1.24–3.0	0.003	1.40	0.82–2.38	0.221
Prior chemotherapy history								
No	130	50	1	—	—	1	—	—
Yes	62	39	2.42	1.59–3.7	<0.001	2.23	1.13–4.40	0.021
Bone pain								
No	160	71	1	—	—	—	—	—
Yes	32	18	1.41	0.84–2.38	0.190	—	—	—
Skeletal-related events								
No	167	75	1	—	—	—	—	—
Yes	25	14	2.01	1.13–3.57	0.017	1.29	0.64–2.59	0.484
**Laboratory data before enzalutamide**								
Hemoglobulin (g/dL)								
<12	63	34	1	—	—	1	—	—
≥12	69	30	0.54	0.33–0.88	0.014	0.61	0.32–1.14	0.123
Missing	60	25	0.56	0.34–0.95	0.030	0.67	0.31–1.47	0.316
Segment WBC (%)								
<60	52	20	1	—	—	1	—	—
≥60	55	35	1.86	1.07–3.22	0.027	2.18	1.16–4.08	0.033
Missing	85	34	1.22	0.7–2.12	0.489	1.30	0.61–2.74	0.594
Platelet count (k/μL)								
<200	70	25	1	—	—	1	—	—
≥200	100	52	1.65	1.02–2.66	0.040	1.15	0.66–1.99	0.628
Missing	22	12	1.47	0.74–2.94	0.269	1.88	0.64–5.56	0.253
Alanine aminotransferase (U/L)								
<16	79	46	1	—	—	1	—	—
≥16	88	33	0.56	0.36–0.88	0.011	0.53	0.31–0.92	0.023
Missing	25	10	0.57	0.29–1.13	0.109	0.56	0.24–1.29	0.172
Alkaline phosphatase (U/L)								
<100	47	22	1	—	—	1	—	—
≥100	19	13	2.60	1.3–5.2	0.007	2.18	1.00–4.77	0.051
Missing	126	54	1.00	0.61–1.64	0.999	1.51	0.77–2.97	0.234
LDH (IU/L)								
<200	21	8	1	—	—	1	—	—
≥200	20	17	4.38	1.89–10.17	0.001	7.46	2.68–20.8	<0.001
Missing	151	64	1.37	0.66–2.87	0.398	1.81	0.76–4.31	0.180
Calcium (mmol/L)								
<2.3	39	23	1	—	—	1	—	—
≥2.3	56	24	0.50	0.28–0.89	0.018	0.58	0.27–1.24	0.162
Missing	97	42	0.58	0.35–0.97	0.037	1.05	0.50–2.21	0.900

## Data Availability

Data available on request due to restrictions of privacy consideration of our institutions. The data presented in this study are available on request from the corresponding author. The data are not publicly available due to the consideration of patients’ privacy.

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
