# Peer review of "High Tumor Burden Predicts Poor Response to Enzalutamide in Metastatic Castration-Resistant Prostate Cancer Patients"

_cancers, 2021, doi:10.3390/cancers13163966_

Round 1

Reviewer 1 Report

The article " High tumor burden predicts poor response to enzalutamide in metastatic castration-resistant prostate cancer patients" authored by Hsieh et is a thorough study and significantly contributes in the field.

Author Response

Thanks for your great effort in reviewing our article.

Reviewer 2 Report

General comments

The objectives of the present study is to evaluate the predictive value of tumor burden on the biochemical response, and radiological response in Taiwanese metastatic castration-resistant prostate cancer (mCRPC) patients receiving enzalutamide in retrospective study. The authors concluded that High tumor burden (HTB) reduced PSA response rate, radiological response rate and progression-free survival duration. As the authors mentioned, the limitation of the present study was retrospective study. Therefore, the detail of the study should be added to the methods and results as specific comments.

Specific comments

Major criticism

  1. Time from diagnosis of mCRPC to administration of enzalutamide should be evaluated in univariable and multivariable analyses of PSA response to enzalutamide in metastatic castration-resistant prostate cancer patients.
  2. History of the treatment with abiraterone should be evaluated in univariable and multivariable analyses of PSA response to enzalutamide in metastatic castration-resistant prostate cancer patients.
  3. The effect of initial hormone therapy for the prostate cancer should be evaluated in univariable and multivariable analyses of PSA response to enzalutamide in metastatic castration-resistant prostate cancer patients.
  4. The authors defined that patients with a HTB were defined as those with either appendicular bony metastasis or visceral metastasis. The remaining patients were considered to be in the low tumor burden (LTB) group. The citation should be added to the sentences. Otherwise, the validity of the definition should be mentioned in the methods.
  5. Early administration of Cabazitaxel has been considered in the treatment for the mCRPC. The treatment strategy of enzalutamide in the authors’ institution and patient selection flow diagram of the present study should be added. For the patients’ selection for the enzalutamide for the treatment, the effect of initial hormonal therapy was not considered?

Author Response

Thank you for giving us the opportunity to modify and thereby improve the manuscript. We have made point-to-point revisions or statements per the reviewer’s comments. All revisions in the manuscript are marked up with “track change” function for reference. We appreciate yours and the referees’ efforts in helping us improve the quality of our paper, and hope this revision be published in Cancers finally.

Our responses to the critique are listed below:

Reviewer 2:

  1. Time from diagnosis of mCRPC to administration of enzalutamide should be evaluated in univariable and multivariable analyses of PSA response to enzalutamide in metastatic castration-resistant prostate cancer patients.

Response: Thanks for your suggestion. We added the parameter, time from mCRPC to the administration of enzalutamide, in Table 1~4. The median time was 4 months without difference between HTB and LTB groups. This variable was used in the logistic regression model for PSA response and radiological response and Cox proportional hazard model for progression-free survival. However, it did not show the significance of prediction value on these three outcomes in the univariable analysis and was reasonably excluded in the multivariable models.

  1. History of the treatment with abiraterone should be evaluated in univariable and multivariable analyses of PSA response to enzalutamide in metastatic castration-resistant prostate cancer patients.

Response: None of our patients in this cohort had a history of the treatment with abiraterone. Based on our patient selection criteria (the new figure as Figure 1) in this study, the patients who ever used abiraterone were totally excluded from the analysis. We would add a clear description of the selection flow in section 2.1.

  1. The effect of initial hormone therapy for the prostate cancer should be evaluated in univariable and multivariable analyses of PSA response to enzalutamide in metastatic castration-resistant prostate cancer patients.

Response: In our study, the effect of initial hormone therapy could be reflected by the time from mPC to mCRPC. To investigate the association between the effect of initial hormone therapy and outcomes, the variable, time from mPC to mCRPC, has been put in the prediction model of PSA response, radiological response, and progression-free survival. It did not show the prediction value on PSA and radiological response but did predict progression-free survival significantly in the univariable analysis. The longer duration from mPC to mCRPC improved progression-free survival (HR = 0.98, 95% CI. 0.97-1.00, p = 0.01). However, it loses the statistical significance of predicting progression in the multivariable analysis (HR = 0.99, 95% CI. 0.98-1.00, p = 0.089).

  1. The authors defined that patients with a HTB were defined as those with either appendicular bony metastasis or visceral metastasis. The remaining patients were considered to be in the low tumor burden (LTB) group. The citation should be added to the sentences. Otherwise, the validity of the definition should be mentioned in the methods.

Response: Thanks for your reminder. The definition of extensive disease from SWOG-8894 study (N Engl J Med 1998; 339:1036-1042) was cited in the section of materials and methods. In addition, the definition was also validated by Riguad et al. (J Urol. 2002 Oct;168(4 Pt 1):1423-6.)

  1. Early administration of Cabazitaxel has been considered in the treatment for the mCRPC. The treatment strategy of enzalutamide in the authors’ institution and patient selection flow diagram of the present study should be added. For the patients’ selection for the enzalutamide for the treatment, the effect of initial hormonal therapy was not considered?

Response: Thanks for your comment. Based on the clinical practice in Taiwan, Cabazitaxel is considered to be used in patients who fail from Docetaxel. Besides, enzalutamide is reimbursed and used for patients with mCRPC prior to or following the use of Docetaxel. Therefore, there were no patients having Cabazitaxel before the use of enzalutamide in our cohort. We will add the above sentence in the Results section. As your recommendation, we also established the patient selection flow diagram (Figure 1) which might be helpful for readers to comprehend the general appearance of our study. There is one important criterion prohibiting us to use reimbursed enzalutamide in Taiwan. In the mCRPC patients, docetaxel was suggested for the patients with both short response duration of ADT (< 12 months) and Gleason grade group of 4 or more. Hence, the median time from mPC to mCRPC was more than 12 months regardless of tumor burden in our study. Hence, we didn’t enroll enough patients who had short duration from mPC to mCRPC in our cohort. The reimbursement issue limits our study to analyze the effect of tumor burden for the patients with poor response to initial hormonal therapy. We would state this in the limitation section.

We trust that with these changes, our manuscript can now be formally accepted for publication in the Journal

Respectfully Your,

Round 2

Reviewer 2 Report

The authors honestly replied to the comments by the reviewer.
I recommend this manuscript to be published to Cancers. Many thanks.

Author Response

Thanks for your helpful review.